# Comparison of Predatory Performance among Three Ladybird Species, *Harmonia axyridis*, *Coccinella septempunctata* and *Hippodamia variegata*, Feeding on Goji Berry Psyllid, *Bactericera gobica*

**DOI:** 10.3390/insects15010019

**Published:** 2023-12-31

**Authors:** Pengxiang Wu, Jia He, Yang Ge, Zhanghui Liu, Runzhi Zhang

**Affiliations:** 1Key Laboratory of Zoological Systematics and Evolution, Institute of Zoology, Chinese Academy of Sciences, Beijing 100101, China; wupengxiang@ioz.ac.cn; 2Institute of Plant Protection, Academy of Ningxia Agriculture and Forestry Science, Yinchuan 750002, China; hejiayc@126.com; 3State Key Laboratory for Quality Ensurance and Sustainable Use of Dao-di Herbs, National Resource Center for Chinese Materia Medica, China Academy of Chinese Medical Sciences, Beijing 100700, China; 13120154491@163.com; 4MOA Key Lab of Pest Monitoring and Green Management, Department of Entomology, China Agricultural University, Beijing 100193, China; 15879774115@163.com; 5College of Life Science, University of Chinese Academy of Sciences, Beijing 100049, China

**Keywords:** functional response, intraspecific interaction, consumption, biological control

## Abstract

**Simple Summary:**

The goji berry, *Lycium barbarum* L., is a prominent indigenous medicinal plant in Northern China. The high nutrient content of fruits may make goji berry plants susceptible to pests. The goji berry psyllid *Bactericera gobica* is an extensively prevalent and devastating pest. The current primary approaches employed for goji berry psyllid prevention involve the utilization of chemical pesticides, which may potentially pose risks to human health and the environment. The advantage of biological control, in contrast to alternative methods, lies in its provision of a cost-effective and sustainable control system with minimal adverse effects. By comparing the predatory performance of three ladybird species preying on psyllids, our study suggests that each species possesses distinct advantages as potential predators of psyllids. Further field studies are necessary to determine the most promising ladybird species for rapid impact through inundative biological control, considering the specific environmental adaptability of each ladybird species. This study is expected to provide evidence supporting the potential of promising ladybird species as an effective biological control agent against goji berry psyllids.

**Abstract:**

The psyllid *Bactericera gobica* is a serious pest in goji berry orchards. The current primary psyllid control methods involve chemical pesticides, which pose potential risks to human health and the environment. The implementation and promotion of biological control agents should receive increased attention as an alternative approach to safeguarding goji berry orchards. To compare the predatory performance of three potential biocontrol agents of psyllids, including *Harmonia axyridis*, *Coccinella septempunctata* and *Hippodamia variegata*, the functional response and intraspecific interactions of adult ladybirds were studied under laboratory conditions. We observed a significantly higher searching efficiency (0.84 ± 0.09) in *H. axyridis* when preying on psyllids compared to *H. variegata* (0.55 ± 0.05), whereas the handling time for psyllids was considerably longer in *H. axyridis* (7.33 ± 0.83 min) than in *H. variegata* (5.67 ± 0.97 min). The impact of intraspecific interactions on *H. variegata* (0.44 ± 0.04) was significantly greater than that on *C. septempunctata* (0.29 ± 0.03), whereas the maximum consumption by *C. septempunctata* (223.35 ± 41.3) significantly exceeded that of *H. variegata* (133.4 ± 26.9). Our study suggests that each of these three ladybird species possesses distinct advantages as a potential predator of psyllids. However, further field studies are required to determine the most promising ladybird species for rapid impact through inundative biological control, taking into consideration the specific environmental adaptability of each ladybird species. The present study is expected to provide evidence that supports the potential of incorporating promising ladybird species as an effective biological control agent in goji berry orchard management programs.

## 1. Introduction

The goji berry, *Lycium barbarum* L. (Solanaceae: Lycium Linn.), is a prominent medicinal plant indigenous to China [1]. The efficacy of goji berries has been extensively investigated and substantiated in terms of their ability to augment energy levels, enhance sleep quality, and confer a diverse array of additional health benefits [2,3]. The high nutrient content of fruits may render goji berry plants vulnerable to pests. The vegetative growth of goji berry is nearly equivalent to its reproductive growth [4]. Deterring pests in goji berry cultivation presents a significant challenge due to the prolonged young fruit period and relatively short periods of fruit expansion and maturity [5]. The reckless employment of chemical control measures not only seldom attains the desired level of efficacy, but also readily engenders significant concerns such as pesticide residue and environmental contamination [6,7,8]. The decline in both the quantity and quality of goji berries can be partially attributed to the frequent occurrence of pests, which is likely influenced by recent climatic factors. The recent frequent observations of the psyllid *Bactericera gobica* Loginova (Hemiptera: Psyllidae), previously known as *Paratrioza sinica* Yang et Li, have been noted in goji berry orchards [9].

The psyllid *B. gobica* is an extensively prevalent and devastating pest infesting goji berries across the northern regions of China, which are major producers of this fruit [10]. The adult psyllids exhibit remarkable jumping abilities and demonstrate strong behavioral tendencies, displaying significant preferences for oviposition and feeding on fresh foliage [11]. The first instar nymphs exhibit heightened levels of activity, whereas nymphs from the second instar onwards tend to settle on leaves and engage in sap-sucking behavior, leading to aberrant leaf development and premature defoliation, and the honeydew secreted by nymphs frequently serves as a catalyst for the occurrence of sooty mold outbreaks [12]. The goji berry psyllid is characterized by its high fecundity, overlapping generations, and exceptional ability to develop resistance against pesticides. It completes four generations annually and overwinters as an adult in the goji berry region of northern China [13].

The primary methods currently employed for goji berry psyllid prevention involve the utilization of chemical pesticides, which may pose potential risks to human health and the environment [14,15]. The excessive utilization of chemicals not only results in the elimination of natural enemies, but also causes the emergence of pesticide-resistant pests, thereby facilitating the resurgence of pest populations [16]. Therefore, greater emphasis should be placed on the promotion of biocontrol agents as a viable alternative for protecting goji berry orchards. The utilization of natural enemies in biological control provides an ecologically sustainable and highly efficient approach to pest management, effectively reducing or mitigating the impact of pests [9]. The advantage of biological control, as opposed to other methods, lies in its provision of a cost-effective and sustainable control system with minimal adverse effects. Bio-control is both safe and environmentally friendly, while also possessing self-dispersing properties [17]. The foraging behavior of ladybird species *Harmonia axyridis* Pallas, *Coccinella septempunctata* Linnaeus, and *Hippodamia Variegata* Goeze (Coleoptera: Coccinellidae) preying on psyllids is frequently observed in goji berry orchards. However, their potential for use as a biocontrol against psyllids, with the most promising ladybird species being used for rapid impact using an inundative biological control, is still unclear. We expect this study to provide evidence supporting the potential of incorporating promising ladybird species as an efficacious biological control agent in goji berry orchard management programs. In this study, we conducted a comparative analysis of their predatory performance, including (1) the functional response and (2) intraspecific interactions of three species of ladybirds preying on psyllids.

## 2. Materials and Methods

### 2.1. Insect Rearing

The colony of goji berry psyllid *B. gobica* was established using individuals collected from the greenhouse at the Institute of Plant Protection, Ningxia Academy of Agro-Forestry Sciences. The psyllids were reared under controlled conditions at 25 ± 2 °C, 65% ± 5% RH and 16:8 h L:D, using goji berry seedlings as their host plants. Our previous investigations indicated that adult ladybirds of three species exhibit a higher predation rate on psyllids compared to other developmental stages [18,19,20], so this study primarily focused on conducting an analysis of adult ladybirds. Adult individuals of three ladybird species, *H. axyridis*, *C. septempunctata* and *H. variegata,* were obtained from the experimental farm of Institute of Plant Protection, Academy of Ningxia Agriculture and Forestry Science, Ningxia, China. The ladybirds were raised alongside fresh psyllids under the same climatic conditions as previously described. Neonates of ladybirds were immediately separated upon hatching to mitigate the occurrence of sibling cannibalism. Plastic containers (16 cm × 22 cm × 8 cm) housed an average of 20 ladybirds belonging to the same species. Adult ladybirds that had recently emerged (<12 h) were selected and subjected to a 24 h period of starvation prior to the commencement of the experiments. During the experiments, ladybirds and fresh psyllids (<12 h) were introduced into Petri dishes (9 cm in diameter) maintained under the aforementioned climatic conditions.

### 2.2. Functional Response

To investigate the functional responses of ladybirds to psyllids, we conducted an experiment involving adult ladybirds from three species (*H. axyridis*, *C. septempunctata* and *H. variegata*), five levels of initial psyllid density (100, 150, 200, 250, and 300 individuals), and four developmental stages of psyllids (eggs, 1st–2nd instar nymphs, 3rd–5th instar nymphs, and adults). The treatments were set up by placing the appropriate number of psyllids on a moistened paper disc in a petri dish, followed by the introduction of a single adult ladybird to the same petri dish. The number of psyllids consumed was examined using a binocular microscope 24 h after introducing the ladybird to the petri dish. The evaluation of two parameters, searching efficiency and handling time, was conducted through the process of equation fitting. Control treatments without the introduction of ladybirds were implemented to account for the inherent mortality of psyllids, thereby adjusting for ladybird consumption of psyllids by eliminating the natural mortality of psyllids. The treatments were replicated 6 times each, with the ladybirds used at a 1:1 male-to-female ratio.

### 2.3. Intraspecific Interactions

The effects of intraspecific interactions on ladybird consumption of psyllids were evaluated by utilizing adult ladybirds from three species (*H. axyridis*, *C. septempunctata* and *H. variegata*) and four developmental stages of psyllids (eggs, 1st–2nd instar nymphs, 3rd–5th instar nymphs, and adults). In a Petri dish, 1, 2, 3, 4 or 5 ladybirds were provided with 100, 200, 300, 400 or 500 psyllids at the same developmental stage, respectively, maintaining a prey-to-predator ratio of 100. Intraspecific interactions among ladybirds for space escalated as the abundance of both ladybirds and psyllids increased in a Petri dish. Parameters of intraspecific interactions and maximum consumption per capita were determined by quantifying the number of psyllids consumed within a 24 h period. Each treatment was replicated 6 times (male: female = 1:1).

### 2.4. Statistical Analysis

#### 2.4.1. Functional Response

The functional responses of ladybirds to psyllids were assessed through a two-stage analysis [21]. The first step involved conducting a cubic logistic regression analysis to determine the shape (type II or type III) of functional responses by examining the proportion of prey consumed (*N*_a_/*N*_0_) as a function of initial density (Equation (1)) [22,23]:(1)$$\frac{N_{a}}{N_{0}}=\frac{\exp(P_{0}+P_{1}N_{0}+P_{2}N_{0}^{2}+P_{3}N_{0}^{3})}{1+\exp(P_{0}+P_{1}N_{0}+P_{2}N_{0}^{2}+P_{3}N_{0}^{3})}$$
where *N*_a_ is the number of psyllids consumed; *N*_0_ is the number of psyllids provided. *P*_0_, *P*_1_, *P*_2_ and *P*_3_ are intercept, linear, quadratic and cubic parameters, respectively. The type of functional response was determined by the direction (positive or negative) of the linear (*P*_1_) and quadratic (*P*_2_) parameters obtained from regression analysis. A type I response is characterized by a linear parameter with *P*_1_ = 0, whereas a type II response exhibits a negative value for *P*_1_, and, finally, a type III response demonstrates a positive value for *P*_1_ along with a negative quadratic parameter (*P*_2_) [24,25]. If the estimates of the linear parameter in the original model did not exhibit statistical significance, we iteratively simplified the model by excluding the cubic term until significant parameters were achieved [26]. The glm function was employed for logistic regression analysis to estimate the values of *P*_0_, *P*_1_, *P*_2_ and *P*_3_ [27].

The second step of the analysis involved employing Holling’s disc equation (Equation (2)) due to the consistent observation of a type II response. This equation was utilized for parameter estimation when quantifying the relationship between psyllid consumption (*N*_a_) and initial psyllid abundance (*N*_0_):(2)$$N_{a}=\frac{aTN_{0}}{1+aT_{h}N_{0}}$$
where *T* is the total time, which, in this case, is 24 h, *a* is the searching efficiency (the area searched per unit time) of ladybird, and *T_h_* is the handling time required to process one psyllid [28]. Parameters *a* and *T_h_* are frequently utilized to estimate disparities in functional response, with both being estimated via non-linear least squares regression using the nls function [29]. The Shapiro–Wilk test was employed to evaluate the assumption of normality, while Levene’s test was utilized to assess the assumption of homogeneity of variances across different developmental stages of psyllids.

The distribution of both *a* and *T_h_* was analyzed following appropriate transformation. The parameters exhibited a positive correlation in all treatments (Pearson’s correlation coefficient *r* = 0.796, *p* < 0.0001, after reciprocal transformation for *T_h_*, Appendix A), within each ladybird species (*r* = 0.869 for *H. axyridis*, *r* = 0.765 for *C. septempunctata*, *r* = 0.855 for *H. variegata*). The differences between ladybird species were investigated for each parameter using ANCOVA, with the other trait as an independent variable and ladybird species as a factor. Prior to analysis, the response variable was transformed using Box–Cox transformation and one outlier was excluded. When the response variable was the searching efficiency (*a*), a significant interaction between handling time (*T_h_*) and ladybird species was observed (*p* of species * *T_h_* = 0.037, Appendix A). The subsequent approach involved the adoption of the same methodology for conducting pairwise comparisons among different species of ladybirds (Appendix A). The same protocol was employed with a reversed order of variables: when the response variable was the handling time (*T_h_*), a significant interaction between searching efficiency (*a*) and ladybird species was also observed (*p* of species * *a* = 0.038, Appendix A). Relevant comparisons were subsequently conducted among pairs of ladybird species (Appendix A).

#### 2.4.2. Intraspecific Interactions

The experiment was conducted to evaluate the impact of intraspecific interactions on ladybirds during predation events. The parameter of intraspecific interactions was estimated through a nonlinear regression analysis (Equation (3); [30]):*y* = *qe*^−*mx*^
(3)
where *y* = ladybird consumption per capita, *e^x^* = initial psyllid abundance, *m* = parameter of intraspecific interactions, *q* = maximum consumption per capita. The model was established to conduct regression analysis on a power function curve.

Similar to the analysis conducted for functional response, the distribution of both parameters *q* and *m* was analyzed following appropriate transformation. The parameters of all species treatments exhibited a positive correlation, both within each species treatment (*r* = 0.820 for *H. axyridis*, *r* = 0.939 for *C. septempunctata*, *r* = 0.764 for *H. variegata*; Appendix A) and across all treatments (Pearson’s correlation coefficient *r* = 0.815, *p* < 0.0001, after log-transformation for *m* and reciprocal transformation for *q*). Differences between species for each trait of *q* and *m* were examined using ANCOVA, with the other trait as the independent variable and ladybird species as a factor. Prior to analysis, the response variable was transformed using Box–Cox transformation and six outliers were removed. When the maximum consumption (*q*) was considered as the response variable, a significant correlation was observed between the intraspecific interaction parameter (*m*) and ladybird species (*p* of species * *m* = 0.015, Appendix A). The subsequent step entailed pairwise comparisons among ladybird species, employing an identical methodology (Appendix A). When considering the intraspecific interaction parameter (*m*) as the response variable, the correlation between the maximum consumption (*q*) and ladybird species remained significant (*p* of species * *q* = 0.048, Appendix A). The next step involved conducting additional comparisons among various pairs of ladybird species (Appendix A).

Descriptive statistics were presented as the mean values, accompanied by their corresponding standard errors of the mean. Disparities between the natural mortality rate and zero were evaluated through a one-sample *t*-test, with statistical significance set at *p* < 0.05. The mean consumption of three ladybird species was assessed for normal distribution and homoscedasticity, followed by analysis using one-way ANOVA with the Tukey HSD test to determine statistical significance at the 5% level. Statistical analyses were performed using the SPSS 20.0 software (IBM, Armonk, NY, USA).

## 3. Results

### 3.1. Functional Response

The natural mortality rates of psyllids were not statistically significant (*t*_119_ = 1.952, *p* = 0.064), as their limited mortality during the experiments rendered the observed rates inconsequential throughout the experimental trials. The parameter estimates were obtained from the logistic model (Equation (1)) for the proportion of psyllids consumed by ladybirds within a 24 h period, in relation to psyllid density (Appendix A). Negative linear parameter *P*_1_ values indicated that the psyllid consumption by all three ladybird species adhered to the type II functional response (Equation (2)). The mean psyllid consumption by *H. axyridis* (86.2 ± 4.3) and *C. septempunctata* (84.2 ± 4.4) exceeded that of *H. variegata* (72.8 ± 3) (*F*_2, 357_ = 3.389, *p* = 0.035). From the perspective of prey consumption, the highest mean number was observed for the 1st–2nd instar nymphs of psyllids (127.7 ± 4.1), followed by the 3rd–5th instar nymphs (94.9 ± 3.0), adults (68.5 ± 1.4), and eggs (33.2 ± 1.4) (*F*_3, 356_ = 212.332, *p* < 0.001).

The psyllid consumption by ladybirds increased proportionally with the number of psyllids supplied for all three ladybird species, reaching a plateau after a certain threshold was reached (Figure 1). Functional response data for psyllid consumption by ladybirds over a 24 h period exhibited a strong fit to Holling’s disc model (*p* < 0.05 for each test), thereby confirming a type II response across all treatments. The searching efficiency (*a*) of *H. axyridis* (0.84 ± 0.09) feeding on psyllids was markedly higher than that of *H. variegata* (0.55 ± 0.05), while the searching efficiency of both species did not differ from that of *C. septempunctata* (0.76 ± 0.10) (Figure 1, Appendix A). However, the handling time for psyllids (*T_h_*) in *H. axyridis* (7.33 ± 0.83 min) was much longer compared to *H. variegata* (5.67 ± 0.97 min), while there was little difference in handling time between *C. septempunctata* (6.42 ± 1.00 min) and the two species (Figure 1, Appendix A).

### 3.2. Intraspecific Interactions

When maintaining a constant psyllid-to-ladybird ratio of 100 in the introduction, the total consumption of psyllids by ladybirds in a Petri dish increased proportionally with the number of introduced ladybirds and psyllids. However, the consumption per ladybird declined as both ladybird and psyllid abundance increased (Figure 2), possibly due to the intraspecific interactions resulting from limited spatial availability. The per capita consumption by *H. variegata* (33.6 ± 1.7) under intraspecific interactions was significantly lower compared to that of *H. axyridis* (40.4 ± 1.9) and *C. septempunctata* (40.5 ± 1.9) (*F*_2, 357_ = 4.763, *p* = 0.009). The per capita ladybird consumption of the 1^st^–2^nd^ instar psyllid nymphs (56.3 ± 1.8) exhibited the highest value among the developmental stages of psyllids, followed by the 3rd–5th instar nymphs (43.1 ± 1.6), adults (38.5 ± 1.4), and eggs (14.7 ± 0.6) (*F*_3, 356_ = 148.819, *p* < 0.001).

The intraspecific interaction equation (Equation (3)) fit all the data remarkably well (*p* < 0.05 for each test). The maximum consumption (*q*) exhibited by *C. septempunctata* (223.35 ± 41.3) significantly surpassed that of *H. variegata* (133.4 ± 26.9), whereas the maximum consumption demonstrated by both species did not differ from that of *H. axyridis* (165.4 ± 37.6) (Figure 2, Appendix A). The impact of intraspecific interactions (*m*) on *H. variegata* (0.44 ± 0.04) was significantly greater compared to *C. septempunctata* (0.29 ± 0.03), while the impact on both species did not differ from that observed in *H. axyridis* (0.33 ± 0.03) (Figure 2, Appendix A).

## 4. Discussion

This study presents a comparative analysis of functional responses to *B. gobica* among *H. axyridis*, *C. septempunctata* and *H. variegata*. The evidence from our laboratory studies demonstrates the potential of ladybirds in suppressing goji berry psyllids. In the study of functional response, the consumption by three adult ladybirds exhibited a curvilinear increase with the abundance of psyllids offered, followed by a saturation point. The disc model proposed by Holling provides a suitable framework for elucidating the consumption patterns of various psyllid species by ladybirds [31,32]. The mean consumption of *H. axyridis* and *C. septempunctata* was significantly higher compared to that of *H. variegata*. The variation in predation capabilities may be attributed to size differences between the predator and its prey. Moreover, the nutritional content of prey has also been acknowledged to impact ladybird consumption [33,34,35]. The selection of prey by predators is driven by the optimization of their intake of essential amino acids [36]. Prey with a higher nitrogen and protein content is typically regarded as a superior dietary choice for predatory ladybirds [37]. Young psyllids may better fulfill the nutritional requirements and intrinsic demands of ladybirds in comparison to mature psyllids. Additionally, the analysis of cuticle thickness and chitin content revealed that adult psyllids possess a more robust and thicker exoskeleton in comparison to nymphs [38,39]. The preference of predators often lies with prey that possess more pliable bodies, as they are more readily digestible and require less energy expenditure during predation [40,41]. The ladybird consumption of psyllid eggs was observed to be the lowest, possibly attributed to the presence of an egg stalk that functions as a protective mechanism against predation by ladybirds [42].

The size of predators also plays a crucial role in shaping their foraging traits, encompassing search efficiency and handling time [43]. The superior efficiency of *H. axyridis* in psyllid search can potentially be attributed to its invasive nature, which confers greater adaptability and enhanced flexibility [44]. The direction and angle of prey are typically perceived with exceptional acuity by predators, facilitating their target localization [45,46]. The rapid prey location recognition based on reference objects in its surroundings is more pronounced in *H. axyridis* compared to *H. variegata* [47]. Flight is the predominant mode of locomotion for *H. axyridis* during foraging, while crawling is the primary means of movement for *H. variegata* [48]. Therefore, the high mobility of *H. axyridis* is likely to enhance its detectability, potentially augmenting its efficacy in psyllid searches. The small-sized *H. variegata* may choose not to engage in hunting when it encounters prey of similar proportions, such as adult psyllids, resulting in a reduced searching efficiency. The decision of whether to engage in predatory behavior towards encountered prey depends on the potential benefits associated with searching for alternative prey items, as predators typically prioritize those that offer the greatest energy yield [49]. However, the handling time required for *H. variegata* to process one psyllid is the shortest once the prey is captured, potentially attributed to its exceptional flexibility resulting from its small size. Specifically, the efficiency of *H. variegata* in handling the stalked eggs of psyllids was notably higher than *H. axyridis* [42]. Overall, three ladybird species each possess distinct advantageous traits in their predation of goji berry psyllids. The larger ladybird *H. axyridis* demonstrates a higher efficiency in searching psyllids, whereas the smaller ladybird *H. variegata* exhibits a shorter handling time.

The investigation into intraspecific interactions revealed a progressive decline in per capita ladybird consumption with increasing ladybird and psyllid abundance. The per capita consumption of ladybirds was negatively impacted by high densities, likely due to the increased likelihood of intraspecific interactions resulting from spatial constraints [50]. The overall consumption sequence remained unchanged in the presence of intraspecific interactions. The intraspecific interaction model for ladybirds has been empirically validated to derive parameters *q* (maximum consumption) and *m* (intraspecific interaction parameter) [51,52]. The maximum consumption of *C. septempunctata* was significantly higher than that of *H. variegata* under intraspecific interactions, possibly due to the limited digestive capacity of *H. variegata* and the phenomenon of predator satiation [53,54,55,56]. The activity of ladybirds significantly increases as a result of consuming an adequate number of psyllids; thus, ladybirds belonging to low-satiation species may allocate a greater amount of additional time to intraspecific interactions [57,58]. The stronger intraspecific interactions observed in smaller *H. variegata* individuals may be attributed to their lower maximum consumption.

This study represents the initial phase in assessing the efficacy of three ladybird species as potential biological control agents against psyllids, focusing on the short-term functional response and intraspecific interaction experiments. The findings suggest that three species of ladybirds each possess unique advantages as potential predators of psyllids. The selection among them may be influenced by the specific ladybird’s environmental adaptability, so it is advisable to choose the dominant ladybird species based on the corresponding environmental conditions. However, the determination of the most promising ladybird species for rapid impact using inundative biological control requires further field studies. Field studies are essential for validating the functional response since the quantitative models developed in controlled laboratory environments seem to have restricted relevance when evaluating foraging capabilities under real-world conditions [59]. This limitation is attributed to variations in predator searching efficiency across diverse environmental contexts [60]. The disparity in outcomes between field and laboratory conditions can be attributed to their spatial complexity, which plays a pivotal role in natural environments and cannot be replicated under simplified laboratory conditions [61]. In fact, the functional responses analyzed in laboratory experiments often reflect the conditions of high pest abundance in the field [62,63]. In addition, the presence of intraspecific interactions can potentially disrupt the quantification of consumption levels, as determined by functional responses. Understanding both psyllid–ladybird and ladybird–ladybird interactions may provide valuable insight into psyllid management, as the existence of intraspecific interactions can potentially disrupt the foraging capacity quantified by functional responses.

During the experiments, we observed an escalation in partially consumed psyllid cadavers as the psyllid abundance escalated, implying that ladybirds demonstrated the capacity to eliminate more psyllids than they could consume due to saturation. The excessive killing and increased consumption indicate the adaptation of predators to environments with abundant food resources and limited food availability, respectively [64,65]. The temperature during the inundative release of ladybirds against psyllids in the field should be considered due to its significant impact on their predatory performance [66]. The consideration of plant characteristics is also valuable for future investigation owing to their impact on the predatory performance of ladybirds [31]. The host plant and its pollen can exert an influence on the palatability and suitability of prey, which may influence the predatory performance [67]. Additionally, considering the prevalence of ladybirds in goji berry orchards and their potential to control psyllids, it is imperative to further investigate the impact of ladybirds on other pests, such as aphids, in the goji berry field. The excessive utilization of chemical pesticides should be minimized, even subsequent to the goji berry fruit harvest, to promote the proliferation of ladybird populations. The implementation of these measures is expected to enhance the sustainability of goji berry production and address safety concerns from a socio-environmental perspective.

## Figures and Tables

**Figure 1 insects-15-00019-f001:**
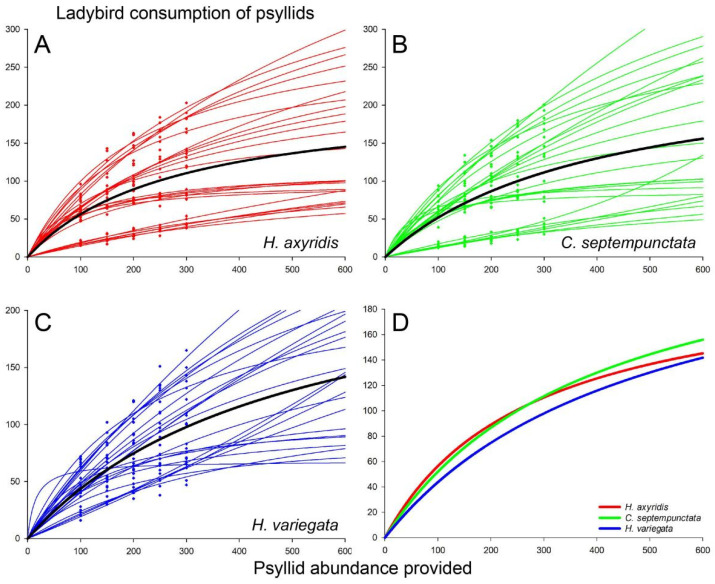
Functional responses of (**A**) *H. axyridis*, (**B**) *C. septempunctata*, and (**C**) *H. variegata* to *B. gobica*. Colored points and curves depict the ladybird consumption of psyllids within 24 h after being introduced to different densities of psyllids. The data for each individual were fitted using the Holling type II functional response function with parameters *a* and *T_h_*. The average values of *a* and *T_h_* in each ladybird species were utilized to generate black curves. (**D**) Schematic representation of psyllid consumption by different ladybird species, with the curves derived from the black curves in (**A**–**C**).

**Figure 2 insects-15-00019-f002:**
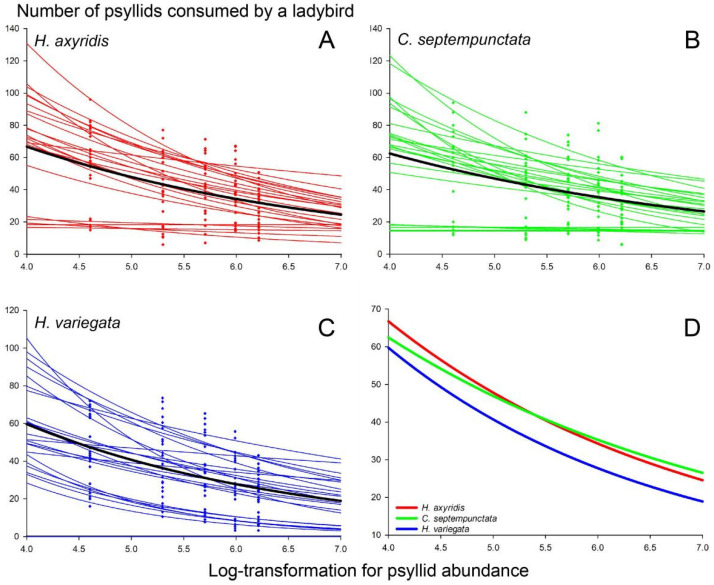
Intraspecific interactions of (**A**) *H. axyridis*, (**B**) *C. septempunctata*, and (**C**) *H. variegata*. The per capita ladybird consumption of psyllids is visually represented through colored points and curves, with a fixed ratio of 100 for different psyllid and ladybird combinations. The data for each test were fitted using the intraspecific interaction function incorporating parameters *q* and *m*. The mean values of *q* and *m* for each ladybird species were employed to generate black curves. (**D**) Schematic representation of per capita consumption of psyllids by each ladybird species, with the curves derived from the black curves in (**A**–**C**).

## Data Availability

The data presented in this study are available on request from the corresponding author.

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
