# Peer review of "Comparison of Predatory Performance among Three Ladybird Species, Harmonia axyridis, Coccinella septempunctata and Hippodamia variegata, Feeding on Goji Berry Psyllid, Bactericera gobica"

_insects, 2023, doi:10.3390/insects15010019_

Round 1

Reviewer 1 Report

Comments and Suggestions for Authors

This is an an interesting and well done study which will be valuable to readers.  However, some key clarifications are needed, and there is also room for reducing the total length of the paper, particularly in the Discussion.

line 93.  Clarify origin of ladybird beetles- commercial insectary? Authors' own rearing colonies?  What are "Dadi eco-cultivation bases of gojiberry"?

l. 111.  Clarify what is meant bby "control treatment" and how control psyllid mortality was used to adjust ladybird consumption values

l. 113.  Clarify "male:female =1:1" ratio of ladybirds? psyllids?

l. 148 - 149.  How were a and Th measured?? (searching efficiency and handling time).

Why weren't a and Th compared directly among species of ladybird through MANOVA instead of ANCOVA?   Advantage of ANCOVA isn't clear.

Overall, it would seem that the presentation and discussion of parameters that may relate to intraspecific interactions is unnecessarily lengthy.  The key results from this paper are the data on functional responses, so the key data are total consumption by ladybirds of the various psyllid stages and the per-capita consumption.  The rest is perhaps theoretically interesting but less relevant, particularly in a petri-dish study.  Once experiments are designed in the field, searching efficiency and handling time can become more relevant.  In the present manuscript, it seems some significant reductions in total material and page number can be obtained.

Comments on the Quality of English Language

Only minor edits needed.

Author Response

This is an an interesting and well done study which will be valuable to readers.  However, some key clarifications are needed, and there is also room for reducing the total length of the paper, particularly in the Discussion.

-Thank you sincerely for the valuable feedback provided by the reviewer. We have diligently incorporated each of the reviewer's suggestions into our manuscript. Notably, we have consolidated two paragraphs discussing intraspecific interaction in the discussion section into a single paragraph. Simultaneously, we have streamlined the entire discussion section by eliminating sentences that are tangential to the main topic and redundant expressions.

line 93.  Clarify origin of ladybird beetles- commercial insectary? Authors' own rearing colonies?  What are "Dadi eco-cultivation bases of gojiberry"?

-The Dadi eco-cultivation bases of goji berry is one of experimental farms of Institute of Plant Protection, Academy of Ningxia Agriculture and Forestry Science.

We have modified it to show the rearing area more clearly: Adult individuals of three ladybird species H. axyridis, C. septempunctata and H. variegata were obtained from the experimental farm of Institute of Plant Protection, Academy of Ningxia Agriculture and Forestry Science, Ningxia, China. 

  1. 111.  Clarify what is meant by "control treatment" and how control psyllid mortality was used to adjust ladybird consumption values

-Modified: Control treatments without the introduction of ladybirds were implemented to account for the inherent mortality of psyllids, thereby adjusting for ladybird consumption of psyllids by eliminating the natural mortality of psyllids.

  1. 113.  Clarify "male:female =1:1" ratio of ladybirds? psyllids?

-Modified: The treatments were replicated 6 times each, with the ladybirds used in a 1:1 male-to-female ratio.

  1. 148 - 149.  How were a and Th measured? (searching efficiency and handling time).

-These two parameters are fitted from equation 2. The reviewer's inquiry appears to be centered around the definition of search efficiency and processing time. This definition has been previously referenced and we have included a citation to that reference.

Poole, A.E.; Stillman, R.A.; Watson, H.K.; Norris, K.J. Searching efficiency and the functional response of a pause-travel forage. Funct. Ecol. 200721, 784–792. https://doi.org/10.1111/j.1365-2435.2007.01288.x.

Why weren't a and Th compared directly among species of ladybird through MANOVA instead of ANCOVA? Advantage of ANCOVA isn't clear.

-The interaction between two parameters in an equation is taken into consideration, ANCOVA can effectively ascertain whether the presence of another parameter influences the relationship between the analyzed parameter and the ladybird species. The MANOVA method, in contrast, is employed for comparing the effects of two or more factors on multiple variables. A crucial assumption underlying this approach is that there exists correlation between the effects of different factors on multiple variables; in other words, one factor's impact will simultaneously affect multiple variables. MANOVA can identify interactions among two or more factors and consider correlations among different variables.

Overall, it would seem that the presentation and discussion of parameters that may relate to intraspecific interactions is unnecessarily lengthy.  The key results from this paper are the data on functional responses, so the key data are total consumption by ladybirds of the various psyllid stages and the per-capita consumption.  The rest is perhaps theoretically interesting but less relevant, particularly in a petri-dish study.  Once experiments are designed in the field, searching efficiency and handling time can become more relevant.  In the present manuscript, it seems some significant reductions in total material and page number can be obtained.

-Thank you very much for the reviewer's comments. We have removed sentences that were relatively redundant and irrelevant. In particular, we have condensed the expression of intraspecific interaction from two paragraphs to one paragraph. Furthermore, we have significantly streamlined the entire discussion section to consist of five concise and clear paragraphs. 

Reviewer 2 Report

Comments and Suggestions for Authors

After reading and analyzing the manuscript entitled "Comparison of predatory performance among three ladybird species, Harmonia axyridis, Coccinella septempunctata and Hippodamia variegata, feeding on goji berry psyllid, Bactericera gobica"by Pengxiang Wu , Jia He , Yang Ge , Zhanghui Liu and Runzhi Zhang, I make the following considerations.

Title: Sugestion, I believe the title could be way more appealing. Make a title about the results, or make an interesting question. As it is, the title is just a summary of the article;

Abstract: When presenting the results, the authors compare the species two by two, so it's confusing to understand which predator performed best. I would think about describing the results differently. Other than that, I think the abstract is very concise and clear;

Keywords: replace words that are already in the title;

Introduction: The authors use much of the introduction to describe the species and the pest, but only one sentence is used to talk about why biocontrol should be chosen. In addition to pesticides, many other forms of control exist; why did the authors choose to delve into biocontrol or instead of botanical insecticides, genetic control, changing cultural practices, etc.?

Linha 74. add (Coleoptera: Coccinellidae);

Material and Methods. item 1.2 Add a phrase summarizing all the parameters evaluated during the experiment;

Discussion: 3. Line 272-303. In geral, the paragraphs are too long, it is hard to ready. I believe there some repeated Information that could be removed, besides that, it would be really interesting to compare the efficacy of these predators to the efficacy of some well known predator of psyllids in China, it would be import in order to really comprehend their feasibility as commercial predators.

Author Response

After reading and analyzing the manuscript entitled "Comparison of predatory performance among three ladybird species, Harmonia axyridis, Coccinella septempunctata and Hippodamia variegata, feeding on goji berry psyllid, Bactericera gobica"by Pengxiang Wu , Jia He , Yang Ge , Zhanghui Liu and Runzhi Zhang, I make the following considerations.

-Thank you very much for the reviewer's comments. We have made modifications one by one according to the reviewer's suggestions. In particular, the first three paragraphs of the discussion section on functional responses have been substantially reduced to one paragraph. At the same time, the whole discussion section has been simplified, deleting sentences that are not closely related to the topic and repetitive expressions.

Title: Suggestion, I believe the title could be way more appealing. Make a title about the results, or make an interesting question. As it is, the title is just a summary of the article;

-Thank you very much for the reviewer's valuable suggestions. Some of the more attention-grabbing titles may have resulted in an exaggerated representation of our manuscript, and we believed that summarizing our experiment in a concise title would effectively convey its essence to the readers.

Abstract: When presenting the results, the authors compare the species two by two, so it's confusing to understand which predator performed best. I would think about describing the results differently. Other than that, I think the abstract is very concise and clear;

-In fact, our study suggests that each of three ladybird species possesses distinct advantages as potential predators of psyllids. The fact remains that various species of ladybugs exhibit distinct advantages in different predation characteristics, with no single species demonstrating a definitive advantage in predation.

Keywords: replace words that are already in the title;

-Modified. Keywords: functional response; intraspecific interaction; consumption; biological control

Introduction: The authors use much of the introduction to describe the species and the pest, but only one sentence is used to talk about why biocontrol should be chosen. In addition to pesticides, many other forms of control exist; why did the authors choose to delve into biocontrol or instead of botanical insecticides, genetic control, changing cultural practices, etc.?

-Thank you for the reviewer's comments. The measures mentioned by the reviewer, such as botanical insecticides, genetic control, changing cultural practices, etc., are not directly relevant to the primary focus of this study on biological control. These aforementioned measures fall outside the scope of this study; therefore, excessive introduction of alternative approaches may divert attention from the manuscript's central theme.

The addition of a statement explaining the rationale behind the utilization of biocontrol and its superiority over alternative measures has been made: The utilization of natural enemies in biological control provides an ecologically sustainable and highly efficient approach to pest management, effectively reducing or mitigating the impact of pests [9]. The advantage of biological control, as opposed to other methods, lies in its provision of a cost-effective and sustainable control system with minimal adverse effects. When managed by experts, bio-control is both safe and environmentally friendly, while also possessing self-dispersing properties [17]. 

Linha 74. add (Coleoptera: Coccinellidae);

-Added.

Material and Methods. item 1.2 Add a phrase summarizing all the parameters evaluated during the experiment;

-Added: The evaluation of two parameters including searching efficiency and handling time was conducted through the process of equation fitting. 

Discussion: 3. Line 272-303. In geral, the paragraphs are too long, it is hard to ready. I believe there some repeated Information that could be removed, besides that, it would be really interesting to compare the efficacy of these predators to the efficacy of some well known predator of psyllids in China, it would be import in order to really comprehend their feasibility as commercial predators.

-Thank you very much for the reviewer's valuable suggestions. We have significantly condensed the initial three paragraphs regarding functional responses into a single paragraph, eliminating any statements that are relatively irrelevant or repetitive. Additionally, we have streamlined the discussion section to consist of five concise and clear paragraphs. 

We believe that this manuscript primarily focuses on comparing the predatory abilities of three ladybug species. The discussion regarding self-comparison is adequate, and introducing comparisons with other predators like lacewings may dilute the main focus of this study since it does not specifically address other psyllid predators.